# Accelerating Sustainable and Economic Development via Scientific Project Risk Management Model of Industrial Facilities

Abdelaal Ahmed Mostafa Ahmed Ragas [1], Alexander Chupin [2,*], Marina Bolsunovskaya [3], Alexander Leksashov [3], Svetlana Shirokova [4] and Svetlana Senotrusova [5]

1. Accounting and Finance Department, United Arab Emirates University (UAE), Al Ain P.O. Box 15551, United Arab Emirates
2. Institute of Foreign Economic Security and Customs Affairs, Peoples' Friendship University of Russia (RUDN University), 6 Miklukho-Maklaya Street, 117198 Moscow, Russia
3. Graduate School of Intelligent Systems and Supercomputing Technologies, Peter the Great St. Petersburg Polytechnic University (SPbPU), 29 Polytechnicheskaya Street, 195251 St. Petersburg, Russia
4. Graduate School of Business Engineering, Peter the Great St. Petersburg Polytechnic University (SPbPU), 29 Polytechnicheskaya Street, 195251 St. Petersburg, Russia
5. Faculty of Public Administration, Lomonosov Moscow State University (MSU), 27-4 Lomonosovsky Prospect, 119234 Moscow, Russia
* Correspondence: chupin-al@rudn.ru; Tel.: +7-977-55-25-618

**Abstract:** This study presents a systemic and causal model of integrated stakeholder risk management of industrial facilities under sustainable development conditions. This model allows us to analyze the main factors of stakeholder influence, namely personnel risks, conflicts, and behavioral economic factors on a scientific project. This method is based on the identification of stakeholders and determining the possibility of the presence in their activities or inaction of personnel risks, conflicts, and behavioral economic factors that can affect the success of the production of industrial facilities, as well as on the calculation of toxicity indicators for each stakeholder. This study presents information technologies for the integrated management of industrial facilities in the context of sustainable development and transition to a circular economy, which, under conditions of uncertainty, allow the manager of an industrial enterprise and his team to implement the methodology of integrated management of industrial facilities in the context of sustainable development and transition to a circular economy to ensure the successful and timely implementation of these projects to meet the needs of stakeholders.

**Keywords:** industry 4.0; sustainable development; integrated risk management; scientific projects; uncertainty conditions; circular economy; personnel risks; conflicts; behavioral economic factors

## 1. Introduction

In the modern world, science and innovation play an increasing role in the development of the state and society, meeting the needs and improving the quality of life of people. Strategic plans and projects for the future, which are developed by the majority of countries' world leaders, are based on knowledge and achievements of science, allowing us to look into the future, to adjust the modern vector of development of the country, and to distribute its resources to ensure the maximum effect in the implementation of certain goals. Science in the XXI century is a strategic resource of the state, the main factor of improving the quality of human capital, the generation of new ideas, the key to building an innovative and competitive economy, and one of the main vectors of industrial development.

Russia has a strong industrial and scientific–technical potential, as well as well-known scientific schools and outstanding scientific achievements, which are concentrated in academic, higher education, and industry sectors. During 2022, there were 950 organizations

engaged in research and development in Russia, of which 48.1% belonged to the state sector of the economy, 37.0% to the entrepreneurial sector, and 14.9% to higher education. In industrial enterprises and organizations engaged in R&D, the number of people performing such work by the end of 2022 was 79.3 thousand, of which 64.5% were researchers, 9.4% were technicians, and 26.1% were auxiliary personnel.

In 2022, the staff share of the total employed population was 0.48%, including researchers at 0.31%. According to Eurostat data, in 2022, the highest share was in Denmark (3.18% and 2.2%), Finland (3.04% and 2.26%), Great Britain (2.29% and 1.68%), and the Netherlands (2.28% and 1.39%); the lowest was in Romania (0.54% and 0.34%), Cyprus (0.87% and 0.62%), Bulgaria (1.09% and 0.71%), and Poland (1.08% and 0.83%). The share of the total R&D expenditure in GDP was 0.47%, including 0.17% at the expense of the state budget [1].

According to the data of 2022, the share of R&D expenditure in the GDP of the EU-27 countries was on average 2.06%. A larger average share of expenditure on research and development was in Sweden—3.4%, Austria—3.16%, Denmark—3.05%, Germany—3.02%, Finland—2.76%, Belgium—2.58%, France—2.19%, but less so in Romania, Latvia, Malta, Cyprus, and Bulgaria (from 0.5% to 0.75%). Among the main reasons negatively affecting the development of the system of science in Russia are the low level of implementation of laws, the lack of a coherent and coordinated policy for the development of the scientific and technological sphere, and insufficient funding of scientific and technological activities. In turn, this leads to the emergence and influence of uncertainty, personnel risks, conflicts, and behavioral economy factors, as well as the loss of scientific potential of projects of industrial facilities in Russia.

In order for a scientific project to be well-planned and implemented at the initiation stage, it is necessary to form an effective project team. Project managers and functional units that are involved in the creation of the project at this stage have to solve a number of specific tasks related to work motivation, conflicts, execution, control, responsibility, communication, power, leadership, etc. This creates favorable conditions for work, helps to overcome the enormous psychological burden that arises during the search, coordination, and implementation of project solutions, and avoids conflicts and stress, which ultimately will affect the appropriate level, quality, and timeliness of the project.

This study conducts conceptual modeling of complex anti-risk management of scientific projects in the conditions of sustainable industrial development and transition to circular economy on the basis of the model "Change Management Iceberg". This conceptual model of complex anti-risk management of scientific projects combines different areas of knowledge and tools related to project management, conflictology, risk management, and behavioral economics. It enables integrated management of risk, conflict, and behavioral economic factors of stakeholders and ensures the successful completion of scientific projects.

Many researchers confirm that about 80% of respondents put the factor of human relations in first place of all factors influencing the successful implementation of any project, so the priority of this area of activity is beyond doubt.

## 2. Literature Review

In particular, the issues of risk management of scientific projects of industrial objects in conditions of sustainable development and transition to circular economy are studied by foreign scientists, including Jing Jia and Zhongtian Li, M. Elisabeth Paté-Cornell, Dmitry O. Reznikov, Nikolay A. Makhutov, Olga N. Yudina, Sónia Almeida Neves, and António Cardoso Marques.

The correlation between the existence of risk management committees and their effective operation has been examined in [1]. The authors measured the readability of risk management disclosures using six different readability indices, namely the Bowg Index; the Flesch Readability Index; the Coleman–Liau Index; the Flesch–Kincaid Grading Level; a simple measure of nonsense language; and the Automated Readability Index. The authors

found that the presence and effectiveness of risk management committees is associated with higher readability of risk management disclosures. The authors used a variety of methods, including the instrumental variable approach, the entropy balancing method, and the dynamic generalized method of moments, to address endogeneity problems. Taken together, the results highlighted the important role of the risk management committee in communicating risk management information.

The limitations and conflicts that are associated with risk management for industrial facilities and the proposed global and explicit approach to numerical safety goals have been addressed in [2]. The use of such numerical goals is part of a more general risk management strategy. Factors such as technical, organizational, ethical, social, legal, and economic factors are involved. In order to prioritize between security measures, the first step is a probabilistic risk analysis, which should not be limited to technical parameters, but should also include organizational factors.

Risk management is one of the key components of the overall process of hazardous production facility management, involving the implementation of a set of measures aimed at reducing the probability of accident scenarios and mitigating their consequences. In [3], the application of the ALARP principle for making management decisions to implement a strategy to reduce the individual risk caused by hazardous production facilities is considered.

A paradigm shift from a linear economy to a circular economy is critical to reducing the burden on the environment and increasing the reliability of primary raw material supplies. Under this new paradigm, governed by the imperatives of "reduce, reuse and recycle", the extraction of primary resources is minimized by extending the life of existing resources and materials. An innovative contribution [4] to this area of research is the empirical confirmation of the role of economic, social, and environmental factors in the transition to a circular economy. To do so, the authors analyzed annual data from 2010 to 2019 for 19 European Union countries using a panel-adjusted standard error estimate, which has been shown to be an appropriate estimator for the characteristics of the data. The circular economy was used as a measure of circular economy. The main results show that the age distribution in a country is a significant predictor of the circular economy.

A review of these works shows that the counter-risk management of scientific projects of industrial facilities under conditions of uncertainty and the transition to a circular economy requires the formation of a project team, and the management of this team is a very complex problem which requires constant attention and improvement of already existing models and methods.

Osamah M.M. Al-Matari et al. (2021) [5] defined and analyzed the principles of forming the creative potential of the project team. In order to improve the effectiveness of the team (team), the Personnel Development Assessment Model (P-CMM (People Capability Maturity Model)) was analyzed, which provides a basis for motivating, recognizing, normalizing, and improving best practices in personnel management. Thanks to it, the head of the enterprise realizes the importance of the employee as an individual and the necessity of his further improvement and development of his creative potential. However, to implement this model in scientific projects, scientists in conditions of limited funding of scientific activity will need to raise a significant amount of their own funds specifically for training and stimulating the development of their abilities.

In [6], cognitive dissonance and causes, indicators of a project manager's emotional burnout, and methods to overcome these factors are considered. According to the results of the research, the essence of emotional burnout syndrome and cognitive dissonance of a project manager is defined, and the symptoms of emotional burnout are established and preventive measures to prevent symptoms of emotional burnout are developed. For scientific projects, these suggestions can be taken as a basis, but taking into account that scientific activity is first of all a creative activity, in this case a project manager should be stress-resistant himself and should be able to help each team member to get out of a negative state.

Scientific projects are no exception to the above because one of their features is the labor intensity of their implementation. The main executors of scientific projects are scientists.

During the planning and implementation of any project, particularly a scientific one, the human factor can have a very large impact, both positive and negative. The customer or investor of the project should pay very close attention to the selection of the project manager and the formation of the project team.

In addition, among all the risks of project activities, one of the main risks is human resources risk, since human resources are the main resources of the project.

At the same time, the problem of risks [7] arising in the course of personnel management reflects the increasing importance of the human factor in project management. Human resource management is based on personnel decisions, which are always made under conditions of full or partial uncertainty [8,9]. The range of alternatives for solving personnel problems and the possible consequences for each alternative are directly proportional to the degree of unpredictability of human behavior.

The complexity of the management of personnel risks is due to the fact that they are associated with human resources, which is based on the nature and essence of personality, which is the most complex object of management [10]. Personnel risks can be divided into two groups, that is, personnel risks and personnel management system risks.

Personnel risks arise from the manifestations of professional, business, and personal qualities of the project personnel and include such types as psychophysiological, personal, communicative, moral, educational, professional, and qualification risks and risks of unreliability.

One of the most important elements of the personnel management system is the management of labor remuneration. Remuneration of labor includes both basic and additional funds, which allow the manager to stimulate the personnel with the appropriate amount of remuneration. The next component of the personnel management system, which requires sufficient attention, is motivation. Any manager who wants to achieve high labor productivity through effective activity of his subordinates should take care to provide them with incentives to work, so the main task of modern management is to create such labor conditions in which the potential of employees will be used in the best way.

It is also necessary to pay attention to the accounting of project personnel, the maintenance of which involves constantly taking into account the nuances of the current legislation of Russia on remuneration and constantly monitoring changes. No less important in the personnel management system is the creation of working conditions for employees, because they are mainly the production environment in which human activity takes place during work. Their condition directly affects the level of a person's ability to work, the results of his work, his state of health, and his attitude towards work. Undoubtedly, any management system includes social development of employees, because social development provides recognition of the importance and necessity of creating a team of like-minded people, whose interests will be directly connected with the interests of stakeholders.

## 3. Methods

To achieve the goal of this study and answer the research questions posed, a hybrid methodology was used, combining a systematic literature review and bibliometric analysis, providing an objective review of knowledge related to the field of risk-averse management of scientific projects of industrial facilities under uncertainty and the transition to a circular economy.

On the way to forming a highly effective project team, each member needs to learn how to solve complex problems collaboratively, and how to work together.

To do this, it is necessary to develop a model for the formation of a highly effective project team (Figure 1).

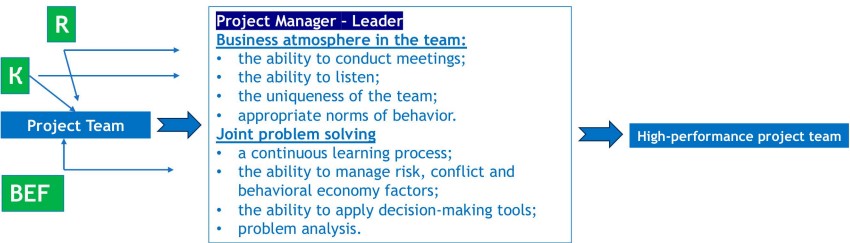

**Figure 1.** Model for building a high-performance project team, where R—risks, K—conflicts, BEF—Behavioral Economics in the Management of Scientific Projects.

Figure 1 shows that the project manager must be a leader and form a highly effective project team that includes experienced, qualified, and capable people. In addition, the project team must exhibit very high performance until it encounters a problem. It is at these moments that the project team shows itself to be a cohesive, high-performing team.

A businesslike atmosphere in a team fosters mutual trust and respect among its members. It also creates the conditions for more productive work and increases productivity. Creating such an environment involves four important elements:

- the ability to conduct each meeting according to its purpose and plan;
- the ability to listen, consisting of the fact that the project manager should have a constructive dialogue with all team members and listen to the opinion of each of them;
- team uniqueness, or the willingness of each team member to work to achieve the goal and obtain a unique and high-quality project product;
- appropriate norms of behavior, that is, personal behavior of each member from the project team which reflects their focus on the result.

With these components in mind, the project manager must be able to apply his or her knowledge, experience, and skills to implement them in the established team.

In addition, the business atmosphere of the team is responsible for two important characteristics of a highly effective project team:

- personal interest of each team member in achieving the overall goal of the project, that is, the overall success of the project should be the subject of their personal and professional growth;
- mutual trust and respect in interpersonal relations of the project team, allowing them to work closely with each other and rely on each other to achieve the project goal.

The ability to work together as a team fosters collaborative problem solving and builds four important qualities in team members:

- a continuous learning process, for the project team to improve its performance during the course of the project by considering both the positives and negatives of its performance;
- the ability to manage conflicts, that is, to use them to find better solutions and avoid damaging relationships in the project team;
- the ability to apply decision-making tools so that all project team members understand all possible ways of making decisions and consciously choose the ones that best fit the specific situation;
- analyzing problems, that is, each of the project team members must understand and use a common problem-solving process that they will follow.

Project management methodology at the present stage considers the project staff as a critical resource for the implementation of any project, including scientific, so the risks associated with personnel take a central place in the overall risk structure of the project [11].

The business environment is highly dynamic with an ever-accelerating rate of change. To remain competitive in a sustainable development industry and transition to a circular economy, companies are actively moving towards project management in order to achieve sustainable business value from their investments [12]. Project management allows organi-

zations to effectively plan, organize, and control the execution of tasks, as well as achieve their goals within a given time frame. With this approach, companies can respond quickly to changes in the external environment and quickly introduce new products or services to the market.

Effective and efficient project management is crucial for the success of any organization. It allows for the effective use of resources, timely completion of projects, and achievement of organizational goals. Additionally, it helps in minimizing risks, optimizing costs, and promoting overall productivity. By considering project management as a strategic competence, organizations can prioritize and align projects with their strategic objectives, ensuring that resources are allocated appropriately and projects are executed efficiently.

Project management in a sustainable development industry also helps to improve communication and coordination between different departments and employees of the company. In doing so, each project has defined goals, tasks, roles, and responsibilities. This avoids duplication of work and conflicts between employees.

In addition, project management promotes innovation and continuous improvement of the company. In the process of project implementation, new ideas emerge, innovative solutions are found, and new products are developed. This helps companies to be ahead of their competitors and successfully cope with the challenges of modern business.

Thus, project management is an essential tool for achieving a competitive advantage and sustainable development in today's business environment.

Every project has stakeholders who are affected by the project or can affect the project positively or negatively.

Stakeholders are people or organizations that have an interest in the successful completion of a project or will be affected by its results. Such stakeholders may include the following:

1. Customer: the person or organization that initiates the project and has the greatest interest in its successful execution.
2. Users: the person or organization that will use or benefit from the results of the project. Their opinions and needs may influence the characteristics and functionality of the project.
3. Management and supervisors: the people responsible for managing and controlling the project. They usually have influence over decision making and risk analysis.
4. Project team: team members who ensure that the various tasks associated with the project are completed. They have an interest in the successful completion of the project and may actively participate in the project.
5. Financial stakeholders: investors, sponsors, banks, or other organizations that fund the project. Their interest is in getting the expected return on investment.
6. Suppliers: organizations or individuals that provide the resources, services, or materials needed to carry out the project. Their work can affect the schedule, quality, and budget of the project.
7. Regulators: government or industry organizations that set the standards or requirements the project must meet. Their role is to supervise and monitor compliance with standards and requirements.

Each stakeholder may have different expectations and goals for the project. Therefore, it is important to communicate and interact with them to understand and meet their needs. This will help create a positive stakeholder influence on the project and achieve its goals.

Stakeholder satisfaction should be identified and managed as one of the project goals. When stakeholders are satisfied, it helps to ensure that the project is successful and achieves its goals.

Stakeholder satisfaction management involves several actions:

1. Identifying stakeholders: All stakeholders who influence or are affected by the project should be identified.

2. Analysis of stakeholder needs and expectations: It is important to understand what stakeholders' needs and expectations of the project are. This may include their business goals, expectations for quality, timing, budget, and other aspects.
3. Communication management: It is important to actively communicate with each stakeholder and take into account their opinions and feedback. This will help prevent or resolve conflicts and reduce the risks of misunderstanding or not meeting expectations.
4. Consideration of stakeholder needs in decision making: When making decisions, it is important to take into account the interests and opinions of stakeholders. They may have valuable ideas or limitations that need to be considered.
5. Stakeholder satisfaction monitoring and evaluation: It is important to regularly assess and monitor stakeholder satisfaction. This will help to identify problems or opportunities for improvement and take appropriate action.

The goal of stakeholder satisfaction management is to create mutually beneficial relationships and to ensure that the interests of all stakeholders are considered and met by the project. This contributes to the success of the project and the achievement of its results [13–16].

Based on the above, it is clear that project activities, in most cases, can be carried out under conditions of uncertainty, in particular risks, conflicts, and behavioral economy factors [17]. Risks are an integral part of any project. They may include financial, technical, operational, or other risks that affect the success of the project. Project risk management involves identifying, analyzing, and monitoring risks and developing strategies to manage them, including prevention or mitigation.

Conflicts can also arise in project activities, especially when different departments or stakeholders with different interests and goals are involved. Managing conflicts in a project involves identifying and resolving conflict situations, as well as finding compromise solutions and supporting cooperation among all participants.

Behavioral economics factors can also affect project activities. Behavioral economics explores how human behavior and decisions may differ from those assumed by economic theory. In projects, this can mean that project participants may make decisions that are not always consistent with rational or economic assumptions. Managing these factors can include motivating participants, communicating, and creating a supportive work environment.

Scientific projects have their own characteristics related to the uncertainty of conducting scientific research. In scientific work, results can be unpredictable and require additional time and resources to achieve the objectives. Managing research projects involves a flexible approach to planning and implementing variables that may arise during the research process.

Thus, understanding and managing uncertainty, risk, conflict, and behavioral economics are important aspects of project activities, including scientific projects.

Therefore, in today's changing environment of the project and its uncertainty, a new methodology of integrated risk management of scientific project conflicts in a behavioral economy is needed.

The success of a project greatly relies on the project manager's ability to effectively manage the team and integrate different methodologies. Convergence, hybridization, and integration of different methodologies have gained significant attention from researchers and practitioners as they offer a holistic approach to project management.

By combining various methodologies, project managers can leverage the strengths of each approach and address the specific needs and requirements of the project and its stakeholders. This integrated approach allows for flexibility, adaptability, and the ability to tailor project management practices to suit different project contexts.

For example, agile methodologies focus on iterative and incremental development, allowing for flexibility and quick responses to changes. Traditional waterfall methodologies, on the other hand, provide a structured and sequential approach that ensures detailed planning and control.

By integrating these methodologies, project managers can benefit from the agility of agile methodologies while still maintaining the structure and control of waterfall. This allows for effective management of the project team and enables the project manager to meet the values and goals of each stakeholder while ensuring the project's success.

In conclusion, the convergence, hybridization, and integration of different methodologies in project management is a valuable approach to ensure success. It provides project managers with the flexibility and adaptability needed to effectively manage their teams and deliver successful outcomes that meet the needs and goals of the project's stakeholders [10,18,19].

In [20], the authors justified the need to develop hybrid methodologies for project, program, and portfolio management. Indeed, in order to effectively manage projects, programs, and portfolios with different lines of business and management methodologies, it would be useful to develop hybrid methodologies. These methodologies can use convergence tools to combine the best practices and approaches of different methodologies, tailored to the specific needs of the project and its stakeholders.

The application of convergence and hybridization techniques requires that the project manager and his team have a sufficient level of competence. They should be familiar with different project management methodologies and be able to adapt them to the specific project. They must also be able to effectively apply convergence tools to form a hybrid methodology that will successfully cope with the requirements of the project and the organizational structure in which it is implemented.

Effective management of human resources and project stakeholders is also an important aspect of successful project implementation. During the project initiation phase, the project manager and his team need to formulate a human resource management policy that will include motivating work, resolving conflicts, establishing clear objectives and control criteria, defining responsibilities, establishing effective communication, and developing systems.

This, in turn, allows managers to create favorable conditions for work, helps to overcome the enormous mental loads arising in the process of search, coordination, and implementation of project decisions, allows managers to avoid conflicts and stresses and to take into account the factors of behavioral economy, which eventually can affect the appropriate level, quality, and timeliness of project implementation [21,22].

Scientific projects are no exception because one of their peculiarities is the labor intensity of its implementation; in particular, their main performers are scientists [23]. During the implementation of a scientific project, scientists are engaged in basic or applied research, collecting and analyzing data, conducting experiments, and developing new concepts and theories. The peculiarities of scientific projects are also related to their non-traditional nature. Such projects usually have specific goals and requirements that are aimed at obtaining new scientific results and establishing new knowledge. In addition, science projects can be carried out within the framework of general programs or initiatives that offer additional constraints and requirements.

The implementation of scientific projects does require a great deal of time, material, labor, and financial resources. Scientists must have access to the necessary infrastructure, laboratory facilities, equipment, and funding to successfully implement a project. They must also have some knowledge and skills to perform research, analyze data, and interpret results.

Despite the complexity and labor intensity, scientific projects play a key role in the development of science and technology. They contribute to the development of new discoveries, innovations, and progress in various fields, and their results can be applied to solve real problems and create new products or services [24].

Thus, understanding the features of scientific projects, their labor-intensive and unconventional nature, as well as ensuring a sufficient level of resources and competence of specialists are important aspects of the successful implementation of scientific projects.

In addition to the project manager and the project team, there are various stakeholders involved in a research project. Stakeholders are individuals or groups who have a vested interest or are affected by the project outcomes. They can include the following:

1. Internal stakeholders: These are individuals or groups within the organization who have a direct interest in the project, such as senior management, department heads, or other teams that are dependent on the project's deliverables.

2. External stakeholders: These are individuals or organizations outside of the project team but have an interest in or influence on the project. They can include clients, customers, suppliers, regulatory agencies, community groups, or any other parties that are impacted by the project.

3. Project sponsors: These are individuals or groups who provide the necessary resources, support, and funding for the project. They have a direct interest in the successful completion of the project and its alignment with the organization's objectives.

4. End-users or beneficiaries: These are the individuals or groups who will ultimately benefit from the project outcomes once it is completed. They can be customers, users, or the general public.

Proper identification, communication, and management of stakeholders are essential for project success. Understanding their interests, needs, and expectations helps the project manager to effectively engage and involve them throughout the project lifecycle, ensuring their positive impact on the project and minimizing any negative influences [25].

In a research project, stakeholders can influence and be influenced by the project. In addition to the project manager, team, and performers, some of the possible stakeholders in science projects may include the following:

Funders: grantors, universities, industrial companies, and other organizations that provide funding for the project. They may influence the project through reporting requirements, restrictions on the use of funds, etc.

The scientific community includes colleagues and other scientific researchers who can influence the project through collaboration, knowledge sharing, peer review, etc.

Regulatory bodies are governmental or international organizations that set rules and standards for scientific research. They can influence the project through requirements for ethics, security, data protection, and other aspects of scientific research.

The public—citizens, foundations, NGOs, and other members of the public can influence the project through their expectations and questions about the ethics and social relevance of the research.

It is important to consider the interests and needs of these stakeholders when planning and implementing research projects, as their influence can affect the performance and sustainability of the project. It is also worth noting that risks, conflicts, and behavioral economy factors may be related to the influence of these stakeholders on the project.

So, based on the above, we see that today there is a need to develop a methodology for integrated risk management of scientific projects in the sustainable industrial development context and the transition to a circular economy.

Project management involves a number of processes that typically include planning, risk assessment, resource management, control, and communication. Each of these processes is performed in order to effectively achieve the project's objectives. Some processes may be completed once at the beginning or at the end of a project, such as planning or results evaluation. Other processes, however, run in parallel and are repeated throughout the project, such as work task management or evaluation of completed work [26].

Integration of project processes plays a key role because it affects the success of the project in achieving its goals. The project manager must have the skills and competencies to effectively integrate project processes. He must ensure alignment between the various processes and objectives of the project, as well as with the highest level of the organization's strategic goals.

On the one hand, the project manager must work with the project sponsor to understand the strategic goals, objectives, and expectations. He or she must ensure that the work

and deliverables of the project are aligned with those goals and objectives. This includes understanding of and alignment with the project portfolio, programs, and other areas of the business.

On the other hand, the project manager is also responsible for ensuring that the project team works together. He or she must create an effective team where each team member understands his or her role and responsibilities. The project manager must also prioritize and focus on what is really important to achieve the project goals.

In both cases, the integration of project processes is critical to ensuring the successful completion of the project.

Some projects are categorized as complex due to various factors such as size, scope, technical requirements, multiple stakeholders, or high levels of uncertainty. These projects often require specialized knowledge, robust planning, and effective management to navigate the complexities involved.

Complex projects typically have the following characteristics:

1.　Multiple interdependencies: Complex projects have numerous tasks and activities that are interrelated, meaning that the success of one task or activity depends on the successful completion of others. Managing these dependencies requires careful coordination and communication.
2.　Uncertainty: Complex projects often involve high levels of uncertainty, such as ambiguous requirements, evolving technologies, or external factors that can impact the project. Flexibility and adaptability become crucial to effectively manage unexpected changes or challenges.
3.　Diverse stakeholders: Complex projects typically involve multiple stakeholders with different interests, priorities, and expectations. Managing these stakeholders and their varying requirements can be challenging but crucial for success.
4.　Technical complexity: Projects that involve complex technologies, specialized expertise, or intricate processes require a deep understanding of the subject matter and the ability to coordinate and integrate different components.

Managing complex projects demands a combination of technical skills, leadership abilities, and effective communication. It requires a clear project plan, rigorous monitoring of progress, proactive risk management, and a flexible approach to adapt to changing circumstances. By recognizing the specific challenges and employing appropriate strategies, project managers can enhance the likelihood of successful outcomes in complex projects.

When considering the complexity of a project, the project manager must take into account all elements solely in order to successfully manage the project:

1.　Multiple parts: Complex projects typically consist of multiple subtasks, phases, or components, each requiring separate attention and resources.
2.　Connections between the parts: In complex projects, the various elements may be interrelated and dependent on each other. Successful project management requires an understanding of these connections as well as the ability to manage them.
3.　Dynamic interactions: In complex projects, interactions between different parts can be dynamic and change over time. The project manager must be prepared to adapt to such changes and make appropriate decisions.
4.　Emergent behavior: Complex projects may exhibit unexpected or unexplained results of interactions between parts. The project manager must be prepared for and flexible to respond to such emergent behavior.

Learning and understanding project complexity helps the project manager effectively plan, coordinate, and control the project, anticipate possible problems, and take timely action to correct them.

## 4. Results

So, in this study, the authors used methods and tools from such knowledge areas of classical project management methodology as project integration, resource and stakeholder

management, as well as the knowledge area of scientific project conflict management proposed in [27], which includes recommendations for the risk-free management of scientific projects in a sustainable industrial development and transition to a circular economy.

Given the specificity of scientific projects, there is a need to integrate the management of risks, conflicts, and behavioral economy factors using the "Change Management Iceberg" model proposed by Kruger [28]. Kruger's opinion emphasizes the importance not only of superficial project management, but also of managing the underlying aspects of the project. Superficial management includes cost, quality, and time control, which are very important for successful project completion. However, in order to achieve real change and achieve project goals, it is also necessary to pay attention to the underlying aspects of project management.

Deep management, according to F. Krueger, includes change and implementation management. This means that the project manager must be able to manage and influence the perceptions and beliefs of team members. It is also necessary to consider the power and political aspects of the project and to be able to allocate authority and resources effectively.

Real change in a project often requires profound changes in the behavior, values, and motivation of all project participants. The project manager must be prepared for these changes and apply strategies that will help influence these underlying aspects.

Ultimately, understanding and addressing the underlying aspects of project management will help the project manager not only achieve project results, but also create real and lasting change in the team and the organization as a whole.

Broadly speaking, management is the active influence on a system or organization to achieve given goals or a desired state. In project management, this is especially important because projects usually have finite goals and limited resources.

The project management process includes such steps as planning, organizing, executing, monitoring, and evaluating. In doing so, the project manager makes goal-oriented decisions and applies certain techniques and methods to achieve the goals.

Management involves many activities, such as defining tasks, allocating resources, leading a team, and controlling and adjusting the process of project implementation. The key skills of a manager are planning, organizing, communicating, decision-making, problem-solving skills, and others.

Targeted influence on the system eliminates randomness and allows you to achieve certain results. However, management also requires flexibility and the ability to adapt to changing conditions and requirements. All of this makes it possible to manage a project effectively and achieve its goals.

Based on the high-performance project team model (Figure 1), risk management, conflict management, and behavioral economy factors must be integrated into one process to ensure the success of the project. This is due to the fact that any modern science addresses these issues (Figure 2).

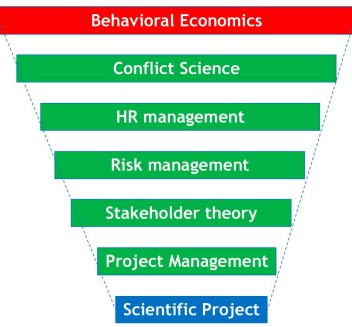

**Figure 2.** Integration of methodologies in scientific project management.

Based on the model of forming a highly effective project team (Figure 1), as well as the integration of different methodologies in scientific project management (Figure 2), it is possible to define the integrated risk management of a scientific project in the conditions of

sustainable industrial development and transition to circular economy and construct the corresponding process (Figure 3).

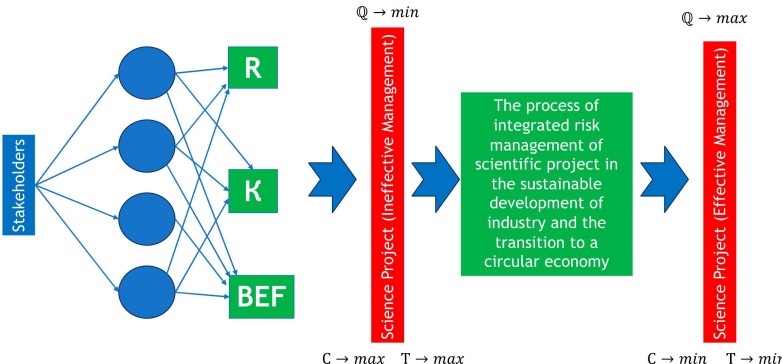

**Figure 3.** The process of integrated risk management of scientific project in the sustainable development of industry and the transition to a circular economy.

Integrated risk management of a scientific project in the sustainable development of industry and the transition to a circular economy is a process that consists in influencing various factors (risks, conflicts, behavioral economy factors) simultaneously, and which aims to continuously restore resources and reduce or eliminate all negative effects in the scientific project, completing it within the approved budget, timeline, and established quality.

The essence of the process of integrated risk management of a scientific project in the sustainable development of industry and the transition to a circular economy lies in the application of a unified process of uncertainty management regardless of the causes of their occurrence.

Based on the process of integrated risk management of a scientific project (Figure 3), we can build a conceptual scheme of integrated risk management of a scientific project in the sustainable development of industry and the transition to a circular economy (Figure 4).

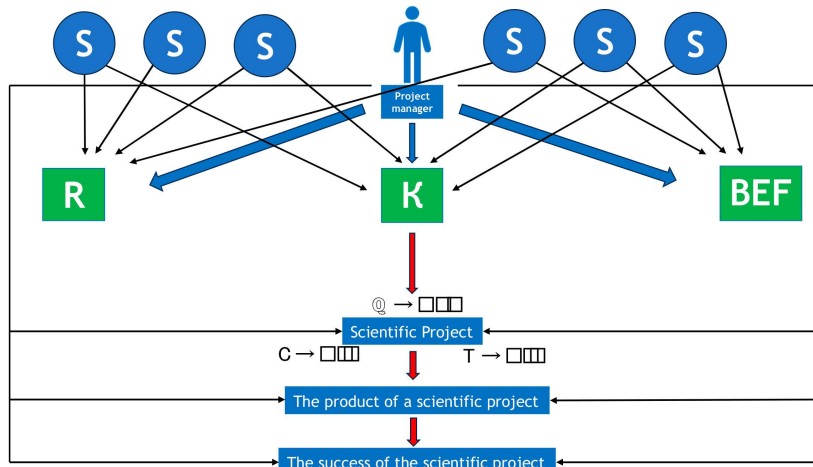

**Figure 4.** The conceptual scheme of integrated risk management of scientific project in the sustainable development of industry and the transition to a circular economy.

Figure 4 shows that risks, conflicts, and behavioral economy factors can arise from the activities of scientific project stakeholders, which should be managed with the help of the integrated risk management of a scientific project in the sustainable development of industry and the transition to a circular economy.

The conceptual model of integrated risk management of a scientific project in the sustainable development of industry and the transition to a circular economy is based on the model of "Change Management Iceberg" [29,30].

This is based on the fact that a feature of the iceberg itself is that there is a larger volume of ice under the water than on the surface of the water. In projection on a scientific project, it can be represented as follows: the environment, stakeholders, some risks and conflicts, and behavioral economy factors are above the water surface, and most of the risks and conflicts are underwater.

Therefore, there is a need to apply integrated risk management of scientific projects under conditions of sustainable industrial development and transition to circular economy in order to identify all hidden personnel risks, conflicts, and behavioral economy factors.

In contrast to existing classical methods of project management, in particular costs, time, and quality, which are on the surface of the iceberg, the conceptual model of the integrated management of a scientific project in the sustainable development of industry and the transition to a circular economy offers to manage the risks and conflicts that are under the water's surface, which are not visible from the outside.

Perception and communication are crucial elements in project management, and it is common for stakeholders to become disengaged or irritated when they feel their needs or concerns are not addressed effectively. Here are a few strategies to improve stakeholder perception and communication:

Understand stakeholder needs: Identify the specific needs and expectations of each stakeholder. Take the time to understand their perspectives, goals, and preferred mode of communication. This will help tailor your communication approach to each stakeholder, ensuring that the information is delivered effectively.

Communicate clearly and proactively: Provide regular project updates and communicate important information in a clear and concise manner. Use different channels (such as meetings, emails, or project management software) to reach stakeholders and keep them informed about project progress, changes, and any issues that may arise.

Use appropriate communication methods: Consider the preferences and communication styles of each stakeholder. Some may prefer face-to-face meetings, while others may prefer written updates or visual presentations. Adjust your communication approach accordingly to ensure that the information is delivered in a format that suits their needs.

Engage stakeholders in decision making: Involve stakeholders in the decision-making process whenever possible. This helps them feel valued and increases their investment in the project. Seeking their input, addressing their concerns, and incorporating their feedback builds trust and enhances their perception of being heard.

Be responsive and proactive: Address stakeholder concerns and questions promptly. Actively listen to their feedback and take appropriate actions to address any issues that arise. Taking a proactive approach to stakeholder engagement shows that you are committed to addressing their needs and demonstrates your dedication to the project's success.

Remember, effective stakeholder perception and communication require ongoing effort and attention. By applying these strategies, you can enhance stakeholder engagement and minimize any frustrations or irritations that may arise during the project implementation process.

Recognizing the emotional state of stakeholders is an important aspect of effective project communication, especially in the case of scientific projects where stakeholders may be scientists or other specialists with a high level of expertise.

When stakeholders do not accept information or even react negatively, this may be due to their emotional state, dissatisfaction, or unconscious biases. Resolving such problems is especially important to maintain and improve relationships with these stakeholders and to ensure the successful progress of the project.

The following approaches can help to recognize the emotional state of stakeholders and improve communication:

1.  Active listening: showing interest in stakeholders' opinions and feelings, finding out their needs and expectations through attentive listening.
2.  Non-verbal cues: observing stakeholders' non-verbal cues and facial expressions to understand their emotional state.

3. Empathy: understanding stakeholders' perspectives and feelings and being able to put yourself in their shoes to better understand their reactions and motivations.
4. Open and honest communication: establishing a trusting relationship with stakeholders where they can comfortably express their emotions and concerns.
5. Adaptability: taking into account the emotional state of stakeholders in the planning and implementation of the project, taking measures to meet their needs and reduce negative emotions.

Recognizing the emotional state of stakeholders and adapting communication accordingly will help establish more effective relationships, reduce conflicts, and increase the chances of a successful research project.

## 5. Conclusions

Thus, the study analyzes the methods of integration of risk management, conflict, and behavioral economics factors. It has been revealed that different sciences offer their own methods and tools for managing risks, conflicts, and behavioral economy factors.

The necessity of building an integrated risk management of scientific projects under conditions of sustainable industrial development and transition to a circular economy has been substantiated. The results of building an integrated approach to managing risks, conflicts, and behavioral economy factors are important for the successful implementation of scientific projects under conditions of sustainable industrial development and the transition to a circular economy.

This study presents a conceptual model of integrated risk management of scientific projects in the conditions of sustainable development and transition to circular economy, which is based on the model of "Iceberg of change management" and is based on the principles of continuous recovery of human resources. From this, it can be seen that both hidden and unhidden risks, conflicts, and factors of behavioral economics can arise from the impacts of the environment and stakeholders. Therefore, there is a need to apply integrated risk management of scientific projects in the conditions of sustainable development and transition to circular economy in order to identify and manage all hidden risks, conflicts, and factors of behavioral economics.

Thus, the modern conditions of functioning of the branch of science, which plays an increasing role in the development of industry of the state, providing the needs and improving the quality of life of people, have been analyzed. It is revealed that science in the XXI century is a strategic resource of industry and the state as a whole, the main factor in improving the quality of human capital, generation of new ideas, and the key to building an innovative, and competitive economy. In view of the current conditions of the transition to a circular economy, it is necessary to ensure the constant recovery of resources in order to find them in the production process for as long as possible, which allows researchers to increase the efficiency of the management process. Also, the analysis of scientific sources regarding the peculiarities of management of scientific projects and the results of research of scientists on the issues of management of labor resources, conflicts, personnel risks, and factors of behavioral economics was carried out. The analysis showed that in projects, human resources risks, conflicts, and factors of behavioral economics, modern science is considered separately, management methods and processes are prescribed differently, and the mutual influence of these threats on each other is not taken into account. For more effective management and sustainable development in the process of transition to a circular economy in scientific projects, the task is to integrate different approaches/processes of management of the above threats into one methodology, which will be universal to manage any uncertainty in the project regardless of the reasons for its sustainable development.

**Author Contributions:** Conceptualization, A.A.M.A.R. and A.C.; methodology, M.B., A.L., S.S. (Svetlana Shirokova) and S.S. (Svetlana Senotrusova); formal analysis, A.A.M.A.R. and A.C.; resources, A.L., S.S. (Svetlana Shirokova); data curation, M.B., A.L. and S.S. (Svetlana Shirokova); writing—original draft preparation, A.C.; writing—review and editing, A.A.M.A.R., A.C. and M.B.; project administration, A.C. All authors have read and agreed to the published version of the manuscript.

**Funding:** The research is funded by the Ministry of Science and Higher Education of the Russian Federation as part of World-class Research Center program: Advanced Digital Technologies (contract No. 075-15-2022-311 dated 20 April 2022).

**Institutional Review Board Statement:** Not applicable.

**Informed Consent Statement:** Not applicable.

**Data Availability Statement:** The data presented in this study are available on request from the corresponding author.

**Acknowledgments:** All authors acknowledge funding support given by the Ministry of Science and Higher Education of the Russian Federation as part of World-class Research Center program: Advanced Digital Technologies (contract No. 075-15-2022-311 dated 20 April 2022).

**Conflicts of Interest:** The authors declare no conflict of interest. The funders had no role in the design of the study; in the collection, analyses, or interpretation of data; in the writing of the manuscript; or in the decision to publish the results.

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
