# Peer review of "Accelerating Sustainable and Economic Development via Scientific Project Risk Management Model of Industrial Facilities"

_sustainability, doi:10.3390/su151712942_

Round 1

Reviewer 1 Report

Review report. Major corrections

The paper "Sustainable Industrial Development and Transition to a Circular Economy Based on the Model of Risk Management of Scientific Projects" explains an organized and causal model of integrated risk management of stakeholders of industrial facilities under circumstances of sustainable development. The model enables analysis of the primary stakeholders' influencing elements, notably risks related to human resources, disputes, and behavioral economic aspects on the scientific project.

Comments on the overall concept: The paper is interesting, and the findings will be useful in real-world situations. The topic is comparatively new in terms of science. According to my anti-plagiarism engine, this collection of authors' works is not comparable to any other works. References used are acceptable. The chapter title "Materials and Methods" is insufficient in this case. Results and analysis ought to be in their own chapter.  The figures are of a suitable caliber. The conclusion is insufficient. The references are pertinent and recent (from the past 5 years or so). Used literature are pertinent. There are not too many self-citations.

Specific comments:

1. The title should be more specific and precise.

2. The abstract is instructive. Please revise statements that begin with "authors have created." It is recommended that you speak in the third person. "It is developed....It was discovered," for example.

3. Keywords are not informative. They are simply repeating what is said in the title.

4. A well-written introduction and a review of the literature. The text is straightforward and easy to read. A comprehensive overview of the literature is provided. The literature review is thorough, well-written, and relevant to the study topic. Nonetheless, the authors should highlight the paper's virtues. What exactly is new and distinct?  What knowledge gap does this paper fill? Please stress this at the conclusion.

5. Line 47: Please avoid abbreviations like R&D.

6. The title Materials & Methods is inappropriate for this section. First and foremost, there is no experimental work and thus no materials synthesis. This section should be renamed. A separate chapter named Methods or Mathematical modeling might be added to discuss methods. The results and comments should be separated into their own chapter.

7. The figures/schemes are of high quality. The manuscript contains no tables. All figures are required.

8. The conclusion must be reworked. Conclusions must only present the work's principal findings. Please emphasize the conclusions you reached.

9. Line 657: “The conceptual modeling of integrated anti-risk management of scientific project under conditions of sustainable industrial development and transition to circular economy based on the model of "Change Management Iceberg" was carried out. This conceptual model of integrated risk management of scientific projects combines different areas of knowledge and tools related to project management, conflictology, risk management and  behavioral economics. It allows the integrated management of risks, conflicts and behavioral economics factors of stakeholders and ensures the successful completion of scientific projects”. This paragraph belongs in the Introduction Chapter rather than the Conclusion Chapter.

10. The listed references are relevant and current (within the last 5 years or so). The used literature is appropriate. There are no excessive self-citations.

English language is good. A native English speaker or a professional proofreader should read and revise the text.

Author Response

Response to Reviewer 1 Comments

Point 1: The title should be more specific and precise.

Response 1: Thank you so much for your comment. We have corrected the title. Please refer to it in the text of the manuscript

Point 2: The abstract is instructive. Please revise statements that begin with "authors have created." It is recommended that you speak in the third person. "It is developed....It was discovered," for example.

Response 2: Thank you so much for your comment. We have corrected it.

Point 3: Keywords are not informative. They are simply repeating what is said in the title.

Response 3: Thank you so much for your comment. We have changed the keywords.

Point 4: A well-written introduction and a review of the literature. The text is straightforward and easy to read. A comprehensive overview of the literature is provided. The literature review is thorough, well-written, and relevant to the study topic. Nonetheless, the authors should highlight the paper's virtues. What exactly is new and distinct?  What knowledge gap does this paper fill? Please stress this at the conclusion.

Response 4: Thank you so much for helping to make our research better. We agree with your comment and in conclusion we have added the virtues of the paper.

Point 5: Line 47: Please avoid abbreviations like R&D.

Response 5: Thank you so much for your comment. We changed that.

Point 6: The title Materials & Methods is inappropriate for this section. First and foremost, there is no experimental work and thus no materials synthesis. This section should be renamed. A separate chapter named Methods or Mathematical modeling might be added to discuss methods. The results and comments should be separated into their own chapter.

Response 6: Thank you so much for reading this carefully. We agree with you that the title "Materials and Methods" is inappropriate. We have changed the name of the section to "Methods".

Point 7: The figures/schemes are of high quality. The manuscript contains no tables. All figures are required.

Response 7: Thank you so much for your comment.

Point 8: The conclusion must be reworked. Conclusions must only present the work's principal findings. Please emphasize the conclusions you reached.

Response 8: Thank you so much for your comment. We have expanded the "Conclusion" section in accordance with your comment.

Point 9: Line 657: “The conceptual modeling of integrated anti-risk management of scientific project under conditions of sustainable industrial development and transition to circular economy based on the model of "Change Management Iceberg" was carried out. This conceptual model of integrated risk management of scientific projects combines different areas of knowledge and tools related to project management, conflictology, risk management and  behavioral economics. It allows the integrated management of risks, conflicts and behavioral economics factors of stakeholders and ensures the successful completion of scientific projects”. This paragraph belongs in the Introduction Chapter rather than the Conclusion Chapter.

Response 9: Thank you very much for your comment. We've corrected.

Point 10: The listed references are relevant and current (within the last 5 years or so). The used literature is appropriate. There are no excessive self-citations.

Response 10: We were very pleased to be able to correct your comments. Thank you very much for being so attentive to our manuscript

Reviewer 2 Report

Manuscript ID sustainability-2521286 Type Article Title Sustainable Industrial Development and Transition to a Circular Economy Based on the Model of Risk Management of Scientific Projects

The authors have developed a systematic and causal model of integrated risk management 18 of stakeholders of industrial facilities under conditions of sustainable development. The model al-19 lows to analyze the main factors of influence of stakeholders, namely human resources risks, con-20 flicts and behavioral economy factors on the scientific project. This method is based on the identifi-21 cation of stakeholders and determining the possibility of human resources risks, conflicts and be-22 havioral economic factors in their activities or inaction that may affect the success of production of 23 industrial facilities, as well as the calculation of toxicity indicators for each stakeholder.

1) The abstract did not show any novelty of the study. The introduction did not show any objectives of the study.

2) There are not enough references in the introduction. For example L52, and L54-L58.

3) some paragraphs are not complete. They are missing the major point like L98.

4) I did not see enough results. Most of the study is based on a literature review.

5) the similarity is high in the section of the literature review.

English must be improved

Author Response

Response to Reviewer 2 Comments

Point 1: The abstract did not show any novelty of the study. The introduction did not show any objectives of the study.

Response 1: Thank you so much for your comment. We adjusted the «Abstract» section and the «Introduction» section of our manuscript.

Point 2: There are not enough references in the introduction. For example L52, and L54-L58. Response 2: Thank you so much for your comment. We have corrected it.

Point 3: Some paragraphs are not complete. They are missing the major point like L98.

Response 3: Thank you so much for your comment. We agree with you.

Point 4,5: 4) I did not see enough results. Most of the study is based on a literature review. 5) the similarity is high in the section of the literature review.

Response 4, 5: Thank you very much for scrutinizing our manuscript. We appreciate your comments. Our study includes a large literature review. It is carefully selected and relevant to the topic of the study, as it is the first time we propose a model we have developed to improve this area of science.

Reviewer 3 Report

It is my pleasure to review the current manuscript titled “Sustainable Industrial Development and Transition to a Circular Economy Based on the Model of Risk Management of Scientific Projects” for the esteemed journal. The manuscript has some merits but there are certain issues which need to be addressed to improve the quality of the manuscript. The authors must consider the following suggestions:

 1. The abstract is well-structured. However, it should further underscore the scientific value added of your paper in your abstract.

2. The novelty of this paper should be further justified by highlighting main contributions of the paper.

3. It would be better if more recent papers are cited.

4. In the methodology section, the model of the study needs to be justified by comparing it with other models.

5. In the results section, the obtained results should be compared with existing studies in the field.

6. Grammar check is required to avoid any possible English errors.

Minor editing of English language required.

Author Response

Response to Reviewer 3 Comments

Point 1: The abstract is well-structured. However, it should further underscore the scientific value added of your paper in your abstract.

Response 1: Thank you so much for your comment. We thank you for helping to make our research better. We have corrected the abstract and slightly expanded the Introduction section.

Point 2: The novelty of this paper should be further justified by highlighting main contributions of the paper.

Response 2: Thank you so much for your comment. We have corrected the «Introduction» section and «Conclusion» section.

Point 3: It would be better if more recent papers are cited.

Response 3: Thank you so much for your comment. We agree with you, which is why we have tried to quote 2023 works in most cases.

Point 4: In the methodology section, the model of the study needs to be justified by comparing it with other models.

Response 4: Thank you so much for such a valuable comment. We have adjusted section 3 "Materials and Methods" and left only "Methods".

Point 5: In the results section, the obtained results should be compared with existing studies in the field.

Response 5: Thank you so much for your comment. The research is devoted to the development of a methodology for integrated risk management of research projects under conditions of sustainable development and transition to a circular economy. The aim of the research is to develop new integrated risk management approaches and methods to solve the scientific and applied problem of research project management in the conditions of sustainable development and transition to circular economy. To define the research problems, the analysis of modern models and methods for managing risks, conflicts and behavioral economics factors in scientific projects in the circular economy was carried out. Based on the analytical review of scientific literature it is determined that our study is unique in this area. Therefore, the results of the study are unique.

Round 2

Reviewer 1 Report

The manuscript is corrected and improved, thereby it can be accepted for publication.

Reviewer 2 Report

I am satisfy with your changes

Some grammar and spelling need too be corrected.